# In Vitro Evaluation of Gastrointestinal Stability of *Pediococcus pentosaceus* Isolated from Fermented Maize and Pearl Millet for Possible Novel Chicken Probiotic Development

**Gifty Ziema Bumbie** [1,2], **Leonardo Abormegah** [2], **Peter Asiedu** [3], **Akua Durowaa Oduro-Owusu** [2], **Kwame Owusu Amoah** [2], **Frederick Danso** [4], **Bernard Bortei Bortieh** [2], **Theresah Nkrumah** [2], **Taha Mohamed Mohamed** [5] **and Zhiru Tang** [1,*]

1. Laboratory for Bio-Feed and Molecular Nutrition, College of Animal Science and Technology, Southwest University, Chongqing 400715, China; giftyziema@gmail.com
2. Council for Scientific and Industrial Research, Animal Research Institute, Accra 20, Ghana; davinciabor@gmail.com (L.A.); sikafuturoa111@gmail.com (A.D.O.-O.); koaari@gmail.com (K.O.A.); bbbortei@yahoo.com (B.B.B.); theresahnkrumah12@gmail.com (T.N.)
3. Department of Animal Production and Health, School of Agricultural and Technology, University of Energy and Natural Resources, Sunyani 214, Ghana; pierro605@gmail.com
4. Council for Scientific and Industrial Research, Oil Palm Research Institute, Kade 74, Ghana; dansotodanso@gmail.com
5. Department of Animal and Fish Production, Faculty of Agriculture (Saba Basha), Alexandria University, Alexandria 21531, Egypt; tahaabdelameed@alexu.edu.eg
* Correspondence: tangzhiru2326@sina.com; Tel.: +86-13996192900

**Abstract:** Research has identified certain bio-based products, such as probiotics, as alternatives to antibiotics for use in animal feed. They are capable of controlling, preventing or minimizing the colonization of the gastrointestinal tract by pathogenic bacteria. To isolate *Pediococcus* spp. and assess its technological properties for possible probiotic development, maize and pearl millet were used. The cereals were steeped and wet milled after 48 h of fermentation. The milled cereals were kneaded into dough for 24 h, after which a 10% slurry was prepared for tenfold serial dilution to enumerate the LAB by employing pour plate techniques using MRS Agar. Based on the cell morphology of the isolated bacteria, eight isolates (four from maize and four from millet) that were selected for identification using MALDI-TOF MS showed that five were *Pediococcus pentosaceus* (*P. pentosaceus)*, one was *Pediococcus acidilactici,* and two did not match any organism. Subsequently, the six isolates were labeled as MZ1, MZ2, MZ3, MZ4 for the maize isolate and MLT5 and MLT7 for the millet isolate. The six *Pediococcus* spp. were assessed in vitro for acid and bile salt tolerance, gastric juice and intestinal fluid tolerance and antibiotic resistance and antimicrobial susceptibility tests were performed, and the survivability rate of the strains was calculated. With regard to the mean count, there was a reduction in log10 CFU/mL under the lower pH conditions and their duration of exposure with regard to time. Among the isolates, no differences were noted at the various periods of exposure (0 h, 1 h, 2 h and 3 h) at pH 4 ($p > 0.05$). However, significant differences were noted at pH 3, 2 and 1 among the isolates ($p < 0.05$). The percentage survival of MZ4 and MLT7 at pH 1 was higher compared to the other isolates at 0 h. Significant differences were observed among the isolated at pH 2, 3 and 4 across the various periods. The mean count of the isolates in gastric juice was similar at 0 and 1 h, but significant differences were noted at 2 and 3 h, where MLT7 was highest ($p < 0.05$). A similar trend was observed for percentage survival. The mean count and the percentage survival of isolates under different concentrations of bile salt were similar. Significant differences were noted among isolates in both mean count and percentage survival when exposed to intestinal fluid ($p < 0.05$). All of the isolates were highly tolerant to the antibiotics tested and possessed antibacterial properties against the selected pathogens. The LABs proved to be good probiotic materials, according to the results obtained. However, the *Pediococcus* strain MLT7 proved to be the LAB of choice; therefore, its molecular identity was verified using the 16S rRNA sequence and was labeled as *Pediococcus pentosaceus* GT001 after it was discovered to have 100% similarity with some strains of *Pediococcus pentosaceus.*

**Keywords:** probiotic; *Pediococcus pentosaceus*; bile salt

## 1. Introduction

The emerging high rate of antimicrobial resistance worldwide has initiated responses from major health regulating bodies such as the World Health Organization [1], Food and Agriculture Organization (FAO), and the World Organization for Animal Health (OIE); therefore, recent strategic plans to curb this menace that poses a serious threat to health and existing drug reserves for treatment of human and animal infections have been drawn up [2]. Meanwhile, research has identified certain bio-based products as potential alternatives to antibiotics when administered through feed that have the ability to treat subclinical infections and prevent the spread of disease in the livestock industry. This novel intervention is becoming the major hope in curtailing the rampant abuse of antibiotics in the industry. On the other hand, scientists in the field of animal nutrition face challenges in finding alternative ways to resolve the misuse of antibiotic growth promoters due to the increasing demand to reduce antibiotic-resistant bacterial pathogens in foods of animal origin. Thus, there is a growing interest in the development of bioactive dietary supplements, such as probiotics, which will provide health benefits to animals when ingested orally in the form of suspensions, powders, capsules or tablets [3]. These alternative bioactive products are expected to provide health benefits to the host by controlling, preventing or minimizing infection by adhering to the intestinal epithelial cells of the host, colonizing it and improving the host's microbiota, protecting it against pathogens and improving its digestive processes [4].

Probiotics are considered viable microorganisms when administered in adequate amounts directly or as feed additives, providing several health benefits to the host, such as the modulation of the microbiome, protection against pathogenic microorganisms, immune modulation, anti-cholesterolemic activity, and their effects on oxidative stress and hypertension [5]. They also improve the quality of meat and decrease noxious gas emissions [6]. To add to the knowledge and possible development of probiotics for broiler chickens using *Pediococcus* spp., an in vitro assessment of the probiotic properties of this lactic acid bacteria (LAB) has become very necessary. *P. pentosaceus* is a promising strain of lactic acid bacteria (LAB) that is receiving more and more attention in experimental studies [7]. It is well known that *P. pentosaceus*, which is isolated from fermented food and found in the digestive systems of both humans and animals, lowers levels of inflammation, liver fat, and obesity. Among the many probiotic advantages of *P. pentosaceus* are its immunological, antioxidant, growth-promoting and cholesterol-lowering qualities [8,9]. In addition to lowering damage to intestinal villi and goblet cells, *P. pentosaceus* dramatically increases complement3 expression and immunoglobin M, indicating immune system stimulation. Additionally, it has broad-spectrum antibacterial properties [8]. In the 1990s, it was established that certain strains of *P. pentosaceus* might be used as probiotics and as bio-promoters of animal growth [10].

## 2. Materials and Methods

### 2.1. Isolation of Pediococcus spp.

Collection of Samples

To ensure the isolation of *Pediococcus pentosaceus* lactic acid bacteria, two types of cereal grains, maize and pearl millet, were acquired from the marketplace in the Greater Accra region of Ghana and transferred to the Council of Scientific and Industrial Research-Animal Research Institute's Microbiology Laboratory. The cereals were steeped separately using clean bowls and allowed to ferment for forty-eight hours.

The maize and pearl millet steeped for forty-eight (48) hours were aseptically wet milled separately using a blender as shown in Figure 1 and kneaded into dough for twenty-four (24) hours, after which a 10% slurry was prepared for tenfold serial dilution. Lactic acid bacteria enumeration was carried out on each cereal slurry by employing the pour plate method using deMan, Rogosa, Sharpe medium (MRS, Oxoid CM361) agar with the pH adjusted to 6.2 using HCl, and 0.1% cycloheximide was included to inhibit the growth of yeast. The inoculated plates were then anaerobically incubated in an anaerobic jar for 48 h at 30 °C using CampyGenTM 2.5 L (Mitsubishi Gas Chemical Company Inc., Tokyo, Japan) to create an anaerobic medium for the period of the incubation [11]. On the appropriate MRS agar plate, colonies of lactic acid bacteria were chosen from a segment of the highest dilution for each cereal, and they were then subcultured into MRS broth (Oxoid CM359; Oxoid Ltd., Basingstoke, Hampshire, UK) and purified on MRS agar (MRS, Oxoid CM361) by streaking repeatedly.

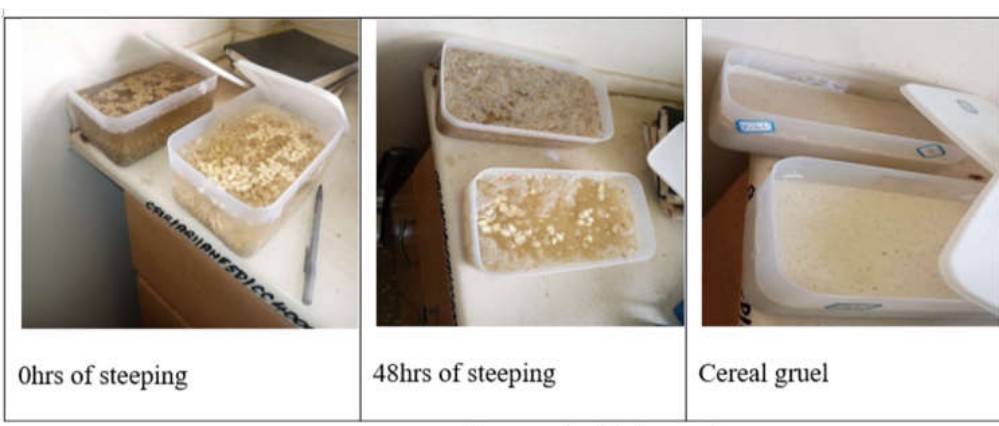

**Figure 1.** Wet fermented cereal.

*2.2. Identification of Pedoicoccus spp.*

2.2.1. Phenotypic Characterization of LAB

Colonial Morphology

Pure colonies of the various isolates obtained from each cereal on the MRS agar plates were observed for their color, size, shape and texture.

Gram Reaction

Gram staining was carried out by emulsifying portions of each purified LAB colony from both cereals in a loop full of PBS on a clean and grease-free microscopic slide using an inoculating loop. The prepared slides were heat-fixed, stained using Gram staining techniques and air dried. To determine the cellular morphology and the Gram reaction of the various isolates on the prepared slides, a light microscope with oil emersion at ×100 magnification was used to examine the slides.

Catalase Reaction

Catalase tests were carried out on all isolates from both cereals by emulsifying portion of the test isolates in three loop fool of freshly prepared 3% hydrogen peroxide solution on clean and grease-free glass. The individual test isolates were then observed for the formation of bubbles or effervescence.

Based on the phenotypic characteristics of all test isolates, all cocci Gram-positive catalyst-negative lactic acid bacteria isolated were grouped into eight (8) categories and coded as follows: MZ1, MZ2, MZ3, MZ4, MLT 5, MLT6, MLT7 and MLT8. MZ1, MZ2, MZ3 and MZ4 were isolated from the maize and MLT 5, MLT6, MLT7 and MLT8 were from the pearl millet. To be able to identify the members of each category group, two representative isolates of each group were identified using MALDI-TOF SM. The representative isolates

presented for identification using MALDI-TOF SM were duplicated and stored at −40 °C in 2 mL scrolled cap tubes containing sterile 50% glycerol for further investigation.

### 2.2.2. Molecular Identification of Pediococcus Strain

#### Extraction of Deoxyribonucleic Acid (DNA)

DNA was isolated according to the protocol provided by Zymo Research (Cat No: D4208T). In detail, cell-cultured samples were emulsified in double-distilled water, taken through a series of centrifugations, washed a number of times and then eluted with an elution buffer, provided in the kit. The concentration of the DNA product was checked using NanoDrop apparatus.

#### 16S rRNA Gene Sequencing

The PCR reaction was carried out using a gradient thermal cycler (Eppendorf, Hamburg, Germany). Specifically, bacterial housekeeping genes (16S rRNA region) were amplified using the universal primers like forward primer -27F (5-AGAGTTTGATCCTGGCTCAG-3) and reverse primer -907R (5′CCGTCAATTCCTTTGAGTTT-3). The total mix was scaled up to a 50 µL reaction mixture and contained 200 µM of dNTP, 0.5 mM of MgCl 2, 200 ng of DNA template, 1.0 U of Taq DNA polymerase and 5 µL of 10 × PCR amplification buffer. The amplification process involved three minutes of initial denaturation at 94 °C, thirty cycles of denaturation at 94 °C for thirty seconds, thirty seconds of primer annealing at 55 °C, and ten minutes of extension at 72 °C. The final amplicons were stored by the machine at a temperature of 4 degrees for further analysis.

#### Agarose Gel Electrophoresis

Subsequently, 10 µL of the reaction mixture was used for 1% Agarose gel electrophoresis, which ran for 45 min at 220 V, and the final product was observed using the gel doc apparatus (Axygen® Gel Documentation Systems, Glendale, AZ, USA). A rapid gel extraction kit (QIAGEN India Pvt. Ltd., Delhi, India) was used to purify the PCR products for Sanger sequencing analysis.

#### Sanger Sequencing

The PCR amplicons were outsourced to Inqaba Biotech, a South African company, for Sanger sequencing. Both forward and reverse sequencing were performed according to the company's in-house protocols. The sequencing was carried out on a Fischer brand ABI 3000XL genetic analyzer (Waltham, MA, USA).

#### Phylogenetic Analysis

Finch TV software was used to visualize the sequences for curation. CLC software was further used for sequence curation and the development of a consensus sequence. The consensus sequence was then submitted for blasting to the National Center for Biotechnology Information (NCBI). The sequences in the NCBI database that were closely comparable to the submitted sequences with very high scores were selected for phylogenetic tree analysis and plotting. The ClustalX MEGA4.0 program was used to achieve multiple sequence alignment, and the Unweighted Pair Group Method with Arithmetic Mean (UPGMA) approach was used to create the phylogenetic tree.

### 2.3. Resistance Tests of Pediococcus spp.

The preparation of the LAB cells was carried out according to Damayanti et al. [8] with little modification. To prepare the isolates for the acid tolerance testing, one of each frozen identified representative of the six groups was re-plated on MRS agar and incubated for 48 h at 30 °C anaerobically. They were then subcultured into a screw-cap tube containing sterile blank MRS broth for 18 to 24 h, after which the screw-cap tube containing the incubated cells of the test organisms was centrifuged at 3000 rpm for 15 min, and the supernatant was discarded. The harvest cell pellets were washed twice using sterile

phosphate-buffered saline (pH 7.3) via centrifugation at 3000 rpm for 15 min. The washed cells were re-suspended into 1 mL PBS for acid tolerance testing [12].

### 2.3.1. Acid Tolerance

To determine the acid tolerance of each representative isolate of the various identified LAB groups, an initial concentration of the test isolates was determined by preparing 0.5 MacFarlane standard equivalent concentration for the various groups using the washed pelleted 1 mL PBS cells of the representatives of the various groups. Using a 1:10 ratio, the varying pH of pH 1, 2, 3 and 4 were used. The survival of cells at various pH levels was determined at 0, 1, 2 and 3 h intervals by pulling 1 mL each at the various interval, serially diluting them using tenfold serial dilution and plating them on MRS agar plates using the plate count technique. The plates were incubated anaerobically at 30 °C for 18 to 24 h. The growth of the cells at various acidities was expressed in log10 CFU/mL. The percentage survival of the cells was calculated as the number of the colonies of the isolate grown on MRS agar divided by the concentration of the initial cells multiplied by 100. The process was triplicated for each group [13]. Percentage survival (%) = log CFU $N_1$/log CFU $N_0$ × 100.

$N_1$ is the count of viable isolates after incubation, and $N_0$ is the initial count of viable isolates.

### 2.3.2. Bile Salt Tolerance

The method used in this study was carried out according to that described by [12] with little modification. To determine the bile salt tolerance of the test isolates, an initial concentration of the test isolates was determined as in the acid tolerance test above. Using a 1:10 ratio, the 0.5 MacFarlane standard equivalent concentration prepared from the representative washed cells of the various groups was used to inoculate the chicken bile salt at different concentrations. That is, 100 μL of the cell concentration was introduced to 900 μL of the chicken bile salt at the following concentrations: 0%, serving as the control, followed by 0.3%, 0.5%, 1.0%, 1.5%, 2.0% and 2.5%. The inoculated bile salt concentrations were incubated at 37 °C for 3 h. The survival of the cells at various chicken bile salt concentrations was determined by serially diluting the various inoculated chicken bile salt concentrations using tenfold serial dilution and plating them on MRS agar plates using the plate count technique. The plates were incubated anaerobically at 30 °C for 18 to 24 h. The growth of cells at various chicken bile salt concentrations was expressed in log10 CFU/mL. The percentage survival of the cells in the chicken bile salts was determined using the calculations of the acid tolerance test above.

### 2.3.3. Artificial Gastric Juice for Gastric Juice Tolerance Testing

To determine the gastric juice tolerance for the prepared test cells of the LAB, artificial gastric juice was prepared by adding 1 g of pepsin to 100 mL of prepared 1 mol/mL HCl (8.33 mL concentrated hydrochloric acid + 91.67 mL sterile $H_2O$) with pH adjusted to 1.5. The prepared mixture was then filtered with a 0.22 μm sterile filter head after thorough mixing. The survival of the cell in the prepared artificial gastric juice was determined by inoculating 9 mLs of the gastric juice with 1 mL of 0.5 MacFarlane standard concentration of the washed cells. The various inoculums were then pulled at 0, 1, 2 and 3 h intervals, where tenfold serial dilution was carried out on each and plated on MRS agar plates using the plate count technique. The plates were anaerobically incubated at 30 °C for 18 to 24 h. The growth of the surviving cells was counted and expressed in log10 CFU/mL [10]. The percentage survival of the cells was also determined, as in the acid tolerance test.

### 2.3.4. Intestinal Fluid Tolerance

Artificial intestinal fluid was prepared by dissolving 6.8 g of $KH_2PO_4$ in 500 mL of sterile distilled water, and the pH was adjusted to 6.8 with 0.4% (*w/w*) NaOH, after which 5 g of trypsin was added, thoroughly mixed and filtered with a 0.22 μm sterile filter head. The survival of cells in the prepared artificial intestinal fluid was determined by inoculating

9 mls of the intestinal fluid with 1 mL of 0.5 MacFarlane standard concentration of the washed cells. The various inoculums were then pulled at two h intervals: 0, 2, 4, 6, 8 and 10 h, where tenfold serial dilution was carried out on each of them before plating on MRS agar plates using the plate count technique. The plates were anaerobically incubated at 30 °C for 18 to 24 h. The growth of the surviving cells was counted and expressed in log10 CFU/mL. The percentage survival of the cells was also determined using the calculations of the acid tolerance test. The process was also triplicated for each group [14].

### 2.3.5. Antibiotics Resistance Testing

To determine the tolerance or resistance ability of the test isolates against some selected G+ antibiotics mostly used in poultry drugs, the identified test LAB isolates representing each group stored at −40 °C were revived by subculturing them anaerobically on MRS agar plates for 48 h at an incubation temperature of 30 °C, after which a few discreet colonies obtained after 48 h were subcultured onto new MRS agar plate for 18 to 24 h. Using the disc diffusion method outlined by Kirby-Bauer [15], a few selected pure LAB colonies of the overnight subcultures were emulsified in sterile physiological saline, and the turbidity was adjusted to the 0.50 MacFarlane standard. Using the surface plating method, MRS agar plates were inoculated with the prepared 0.50 MacFarlane standard test LAB isolates using swab sticks, and the selected antibiotic discs (CAS-POS, Abtek Biologicals Ltd., Liverpool, UK) were placed on the inoculated agar plates and incubated anaerobically at 30 °C. The absence or presence of a zone of inhibition was determined after 18 to 24 h of incubation.

### 2.3.6. Antimicrobial Activity against Three Selected Pathogens

The antimicrobial properties of the test LAB isolates were determined using the agar overlay method described by Ayeni et al. [16] with little modification. To determine the inhibition potentials of the test LABs, a 0.5 MacFarlane standard concentration equivalent to the test LABs was prepared using overnight test LAB isolates and PBS. Using a microliter pipette, two 0.02 mL drops of 0.5 MacFarlane standard concentration equivalent of the PBS prepared test LABs were transferred onto the surface of a sterile blank MRS agar plate at a good distance away from the edge of the plates and allowed to dry. The inoculated plates were incubated anaerobically at 30 °C for 24 h. The 24 h incubated plates showing growth of the tested LABs were overlaid with 10 mL nutrient agar (0.7% agar-agar) containing 0.2 mL of 0.5 MacFarlane standard equivalent PBS prepared overnight from the cultured test pathogens. *Escherichia coli* and *Salmonella typhimurium* were incubated aerobically at 37 °C for 24 h. The overnight-incubated plates were examined for a clear zone of inhibition around the drops of the LAB, and the clear zones were measured.

### 2.4. Statistical Analysis

All of the data were statistically analyzed using Minitab® version 18.1 (Minitab version 18). One-way analysis of variance (ANOVA) was used to evaluate the descriptive quantitative graphs at the 0.05 significance level. Bonferroni post hoc tests were also used to detect individual group differences to ensure reliable comparisons while the viability percentage was calculated by dividing the total survival cell after incubation (log10 cfu/mL) with the total initial viable cells (log10 cfu/mL) and multiplied by 100%. Tukey's test was also used to examine differences in treatment means, with statistical significance set at $p < 0.05$.

## 3. Results

### 3.1. Isolation of Probiotic Potential LAB from Fermented Cereal Gruel

For the isolation of LABs with probiotic potential from fermented cereal gruel, a total of 55 lactic acid bacteria (21 from maize and 34 from millet) were isolated from the fermented cereal gruel. The cell morphology of the isolated bacteria was identified as cocci (33), bacilli (19), coccobacilli (2) and yeast (1). Eight cocci-shaped isolates from each cereal selected for catalase and Gram reaction test were all found to be catalase-negative and Gram-positive.

### 3.2. Phenotypic Characterization and Grouping of the LAB Isolates

Table 1 below shows the cell morphology, Gram reaction and catalase test results of the test isolates obtained after the 48 h fermentation of both maize and pearl millet.

**Table 1.** Phenotypic characteristics of the LAB isolates.

| Isolate's Group ID | Colonial Morphology | Catalase Test | Gram Staining | Cell Morphology |
|---|---|---|---|---|
| MZ 1 | Small, white and smooth | − | + | Cocci in clusters |
| MZ 2 | Medium and off-white | − | + | Cocci in pairs |
| MZ 3 | Off-white and smooth | − | + | Cocci in pairs |
| MZ 4 | Medium white and shiny | − | + | Short chains cocci |
| MLT 8 | White, small and shiny | − | + | Cocci, pairs and tetrad |
| MLT5 | Off-white and small | − | + | Cocci in clusters |
| MLT6 | White and mucoid | − | + | Cocci, single and pairs |
| MLT 7 | White and smooth | − | + | Cocci in clusters |

MZ—maize; MLT—pearl millet.

### 3.3. Identification of Each LAB Isolate Group Using MALDI-TOF SM

Based on their phenotypic characteristics, the isolates were placed into eight (8) groups and coded MZ1, MZ2, MZ3, MZ4, MLT5, MLT 6, MLT7 and MLT8. The results obtained from the two representatives identified from the various groups via MALDI-TOF SM indicated MZ1, MZ2, MZ3, MLT 5 and MLT7 as strains of *Pediococus pentosaceus*, while one representative of the group, MZ4, was identified as *Pediococus acidilactitci*. Two representatives (MLT6 and MLT8) from the groups did not match any organism as shown in Figure 2. MZ isolates are from maize while MLT isolates are from millet.

**Figure 2.** MALDI-TOF MS identification results.

Following a 48-h incubation period, Figure 3 displays the colonial morphology of *Pediococcus pentosaceus* GT001 (MLT7) in addition to it visible cell morphology under a microscope.

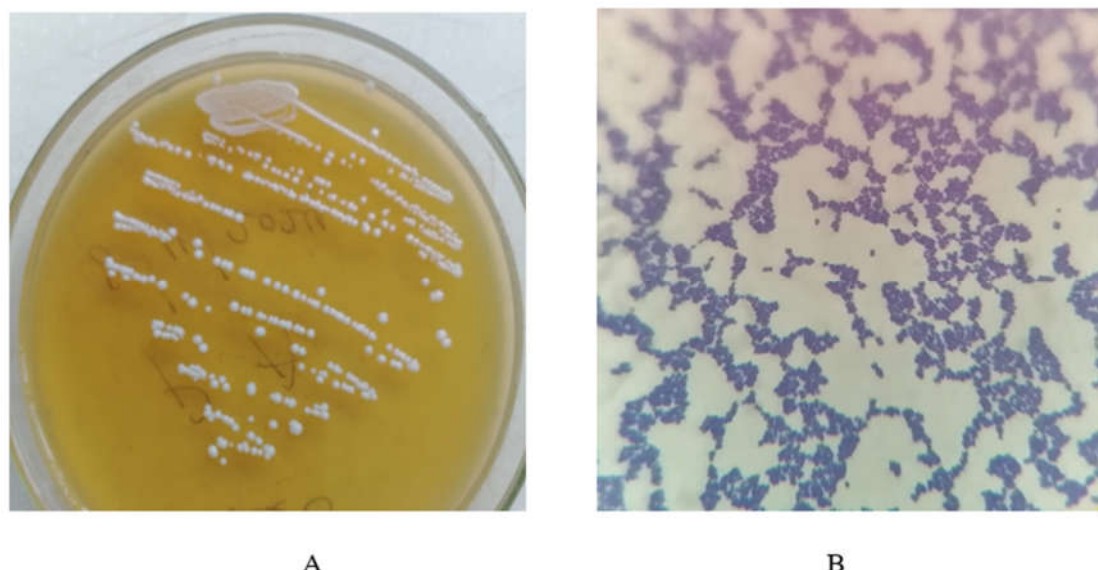

**Figure 3.** The colonial and cell morphology of *Pediococcus pentosaceus* GT001. (**A**) Strain GT001 observed by the eye; (**B**) strain GT001 observed under the microscope.

*3.4. Technological Properties Assay of LAB Isolates In Vitro*

3.4.1. Acid Tolerance

The initial concentration of the test LAB isolates was estimated to be 8.15 log10 CFU/mL; however, the results obtained indicated a reduction of log10 CFU/mL of the test LAB isolates after their exposure to acids with various hydrogen ion concentrations. The decline in their mean concentration was observed further in acids with lower pH—3, 2 and 1. A similar trend was observed in the time duration of their exposure. The initial values recorded at 0 h for pH 4 were similar to that of the initial but also declined with time. The decrease was noted more at lower pH and less exposure with regard to time—0, 1, 2 and 3 h. Among the isolates, no significant differences were noted at the various periods of exposure (0 h, 1 h, 2 h and 3 h) to pH 4 ($p > 0.05$). However, significant differences were noted at pH 3, 2 and 1 among the isolates ($p < 0.05$). The mean counts recorded in relation to the test isolates during their time of exposure and acid pH are shown in Figure 4.

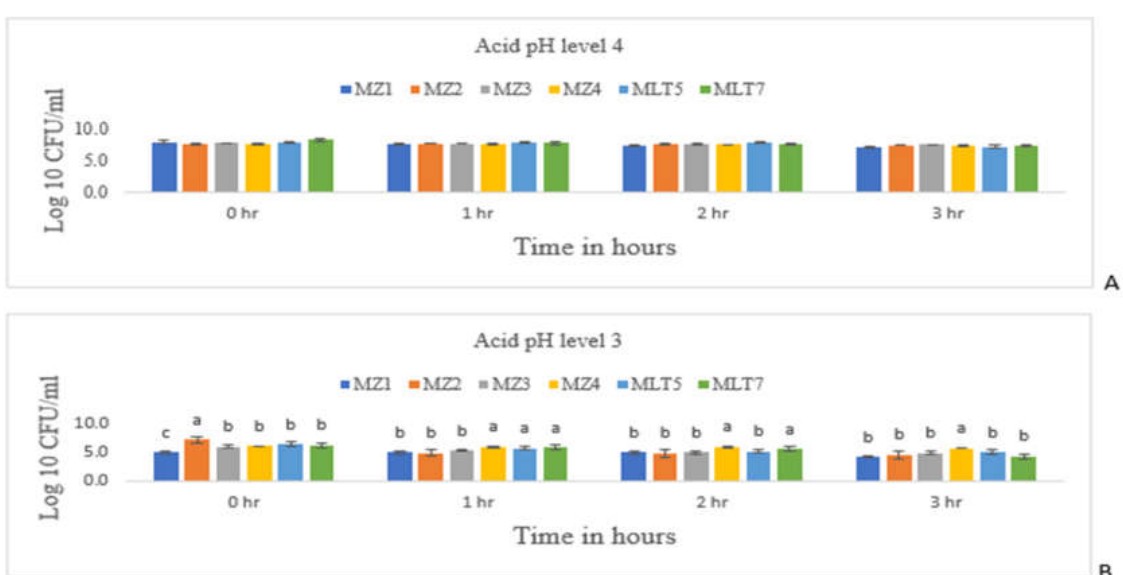

**Figure 4.** *Cont*.

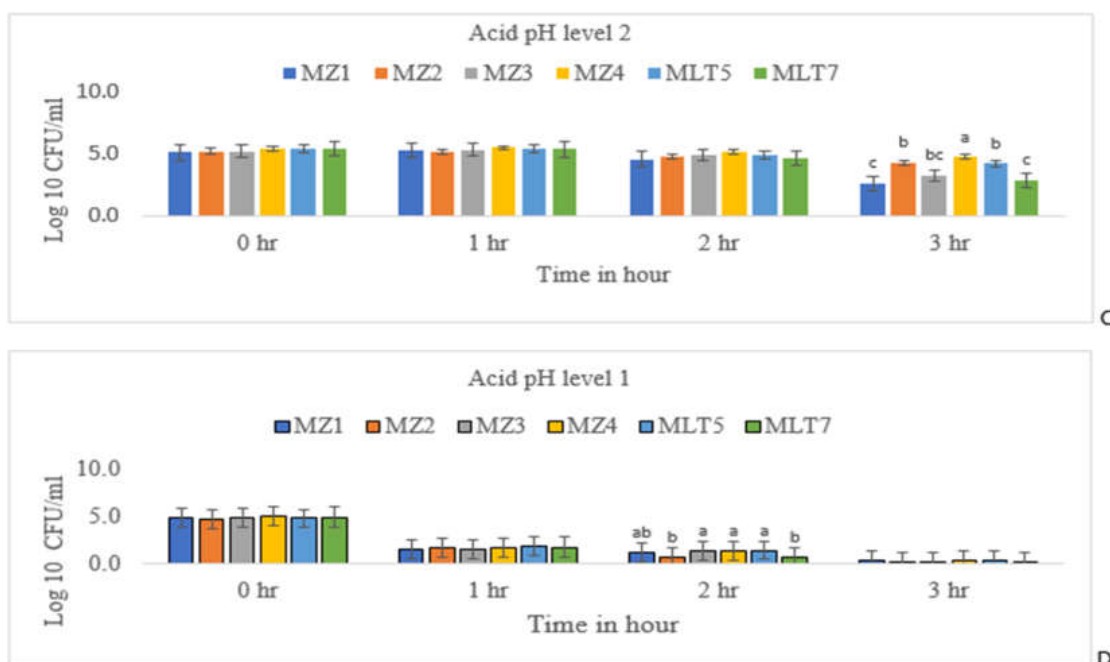

**Figure 4.** Mean count of LAB isolates exposed to different pH: (**A**) pH 4, (**B**) pH 3, (**C**) pH 2 and (**D**) pH 1. [a, b, c] Means in the same row with different superscripts differ significantly ($p < 0.05$). Means in the same row without superscripts are similar ($p > 0.05$).

3.4.2. Percentage Survival Rate of Isolates in Varying Levels of pH

The results in Table 2 show the effects of different pH levels on the percentage survival of the isolates. The percentage survival of isolate MZ4 at pH 1 at 0 h was similar to isolate MLT7, but was significantly higher compared to isolates MZ1, MZ2, MZ3 and MLT5 ($p = 0.001$). The percentage survival of the isolates at pH 1 at 1, 2 and 3 h did not differ significantly among the isolates. The percentage survival at pH 2 at 0 h was similar among the isolates ($p = 0.519$), but numerically, MLT7 and MZ1 were recorded as having the highest values, followed by MT5 and MZ4 and then MZ3 and MZ2. Similarly, there was no significant differences among the isolates at pH 2 at 1 h ($p = 0.613$). A significant difference was observed at pH 2 at 2 h ($p = 0.001$). MZ4 was highest, followed by MZ3 and MLT5, MZ2 and MZ1 and MLT7, which was recorded as having the least. Additionally, MZ4 was significantly higher at pH 2 at 3 h compared to the other isolates. The percentage survival of the isolates varied significantly ($p = 0.001$) at pH 3 at all the periods (0 h, 1 h, 2 h and 3 h). MZ2 was significantly higher compared to the other isolates at 0 h, but the values obtained were similar among MZ1, MZ3, MZ4, MLT5 and MLT7. At 1 h, MLT7, MLT5 and MZ4 were similar but were significantly higher than isolates MZ1, MZ2 and MZ3. The percentage survival of isolates at 2 h was similar among MZ1, MZ2, MZ3 and MLT5, but they were significantly lower compared to MZ4 and MLT7. MZ4 was significantly higher compared to the other isolates. There was significant variation among the isolates at pH 3 at 3 h. MZ1, MZ2, MZ3, MLT5 and MLT 7 were similar, but they were significantly lower compared to MZ4 ($p = 0.001$). Additionally, there was a significant variation in the percentage survival among the isolates at pH 4 at all periods (0 h, 1 h, 2 h and 3 h). MLT7 was recorded as having the highest significant value compared to its counterparts ($p = 0.001$), but similar values were observed among MZ1, MZ2, MZ3, MZ4 and MLT5 at 0 h. The percentage survival of isolates was similar between MLT5 and MLT7 but MLT5 was significantly higher compared to isolates MZ1, MZ2, MZ3 and MZ4 at 1 h. At 2 h, MLT5 was recorded as having the highest significant value, followed by MLT7, MZ2, MZ3, MZ4 and then MZ1, but MZ4 was similar to MZ1. Isolate MLT7 was similar to isolates MZ2 and MZ3 but showed significant variation from isolates MZ1, MZ4 and MLT5 at 3 h, as recorded in Table 2.

**Table 2.** The percentage survival of the test isolates in varying pH concentrations with respect to time.

| pH 1 | MZ1 | MZ2 | MZ3 | MZ4 | MLT5 | MLT7 | SEM | *p*-Value |
|------|-----|-----|-----|-----|------|------|-----|-----------|
| 0 h | 0.04 [bc] | 0.03 [c] | 0.04 [bc] | 0.06 [a] | 0.04 [bc] | 0.05 [ab] | 0.003 | <0.01 |
| 1 h | 0.00002 | 0.00003 | 0.00002 | 0.00003 | 0.00004 | 0.00003 | 0.00001 | 0.126 |
| 2 h | 0.000008 | 0.000003 | 0.00001 | 0.00002 | 0.000013 | 0.000003 | 0.00001 | 0.074 |
| 3 h | 0.000001 | 0.000001 | 0.000001 | 0.000002 | 0.000001 | 0.000002 | 0.00000 | 0.653 |
| **pH 2** | | | | | | | | |
| 0 h | 0.23 | 0.09 | 0.11 | 0.18 | 0.18 | 0.23 | 0.06 | 0.519 |
| 1 h | 0.130 | 0.44 | 0.14 | 0.20 | 0.16 | 0.15 | 0.14 | 0.613 |
| 2 h | 0.02 [d] | 0.04 [c] | 0.05 [b] | 0.10 [a] | 0.05 [b] | 0.03 [d] | 0.0018 | <0.01 |
| 3 h | 0.0003 [d] | 0.01 [c] | 0.0006 [d] | 0.04 [a] | 0.01 [c] | 0.02 [b] | 0.0005 | <0.01 |
| **pH 3** | | | | | | | | |
| 0 h | 0.07 [b] | 11.47 [a] | 0.670 [b] | 0.89 [b] | 2.44 [b] | 1.33 [b] | 0.66 | <0.01 |
| 1 h | 0.07 [b] | 0.06 [b] | 0.16 [b] | 0.67 [a] | 0.44 [a] | 0.53 [a] | 0.05 | <0.01 |
| 2 h | 0.07 [c] | 0.05 [c] | 0.07 [c] | 0.67 [a] | 0.11 [c] | 0.24 [b] | 0.02 | <0.01 |
| 3 h | 0.01 [b] | 0.03 [b] | 0.07 [b] | 0.41 [a] | 0.05 [b] | 0.16 [b] | 0.03 | <0.01 |
| **pH 4** | | | | | | | | |
| 0 h | 60.00 [b] | 28.76 [b] | 46.67 [b] | 28.89 [b] | 53.34 [b] | 117.78 [a] | 8.17 | <0.01 |
| 1 h | 25.74 [c] | 32.22 [bc] | 34.22 [bc] | 28.45 [bc] | 45.78 [a] | 37.11 [ab] | 1.98 | <0.01 |
| 2 h | 16.00 [c] | 28.44 [b] | 26.22 [b] | 23.78 [bc] | 43.56 [a] | 29.78 [b] | 1.98 | <0.01 |
| 3 h | 8.00 [d] | 19.78 [ab] | 22.67 [ab] | 16.00 [bc] | 11.56 [cd] | 24.86 [a] | 1.45 | <0.01 |

Maize. MLT—pearl millet; SEM—standard error of mean. [a, b, c, d] Means in the same row with different superscripts differ significantly ($p < 0.05$). Means in the same row without superscripts are similar ($p > 0.05$).

### 3.4.3. Gastric Juice Tolerance

The results obtained from this test also indicated deviation from the initial concentration of the test LAB isolates, which was estimated to be 8.15 log10 CFU/mL to 7.3 log10 CFU/mL and 7.1 log10 CFU/mL at 0 h in all the test isolates. No significant differences were observed among isolates at 0 h and 1 h ($p > 0.05$), but the mean concentration further decreased to 5.3, 5.1 to 5.0 log10 CFU/mL within an hour, 4.1, 3.2, 3.1 and 3.0 log10 CFU/mL, respectively at 2 h. Similar reduction patterns were recorded in log10 CFU/mL for all the test LAB isolates after their exposure to the gastric juice except in the LAB MLT7 and MZ1, which recorded a high tolerance ability to the gastric juice at the 2nd h, while MLT7 and MZ4 were recorded as having high tolerance abilities at the 3rd h ($p < 0.05$), as shown in Figure 5.

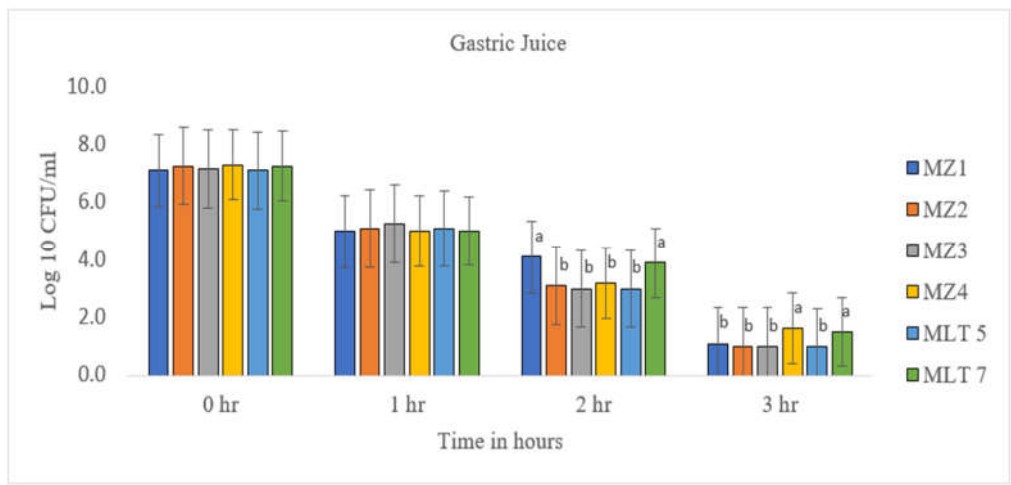

**Figure 5.** Decrease in the mean count of the test LAB isolates exposed to gastric juice of different periods. [a, b] Means in the same row with different superscripts differ significantly ($p < 0.05$). Means in the same row without superscripts are similar ($p > 0.05$).

### 3.4.4. Percentage Survival of Isolates against Gastric Juice

Table 3 shows the effect of gastric juice on the percentage survival of the isolates. The percentage survivability of isolates in gastric juice at 0 h and 1 h did not show any significant variation, but MLT7 was recorded as having the highest value ($p = 0.299$) at 0 h. MLT7 and MZ1 were similar but significantly higher ($p = 0.001$) than that of isolates MZ2, MZ3, MZ3 and MLT5 at 2 h. Percentage survival was similar among LAB isolates MZ2, MZ3, MZ3 and MLT5 at 2 h. At 3 h, MLT7 recorded the highest significant value compared to the other isolates. MZ1, MZ2, MZ3 and MLT5 were similar but significantly lower than MZ4 at 3 h.

**Table 3.** Effect of gastric juice on percentage survival of the isolates.

|  | MZ1 | MZ2 | MZ3 | MZ4 | MLT5 | MLT7 | SEM | *p*-Value |
|---|---|---|---|---|---|---|---|---|
| **Gastric Juice (%)** | | | | | | | | |
| 0 h | 8.89 | 13.33 | 11.11 | 13.33 | 8.89 | 17.78 | 2.87 | 0.299 |
| 1 h | 0.067 | 0.089 | 0.1333 | 0.067 | 0.089 | 0.067 | 0.0202 | 0.223 |
| 2 h | 0.009 [a] | 0.008 [b] | 0.0007 [b] | 0.009 [b] | 0.0007 [b] | 0.0007 [a] | 0.0008 | <0.01 |
| 3 h | 0.000008 [c] | 0.000007 [c] | 0.000007 [c] | 0.00001 [b] | 0.000007 [c] | 0.00007 [a] | 0.000000 | <0.01 |

MZ—Miaze. MLT—Pearl Millet; SEM—Standard Error of Mean. [a, b, c] Means in the same row with different superscripts differ significantly ($p < 0.05$). Means in the same row without superscripts are similar ($p > 0.05$).

### 3.4.5. Bile Salt Toreance

The result obtained shows slight decrease in the mean concentration of the test LAB isolates exposed to the bile salt with the varying concentration. However, the effect was not significant ($p > 0.05$). Figure 6 below shows the various mean count in log10 CFU/mL of the test LAB isolates exposed to different concentrations of bile.

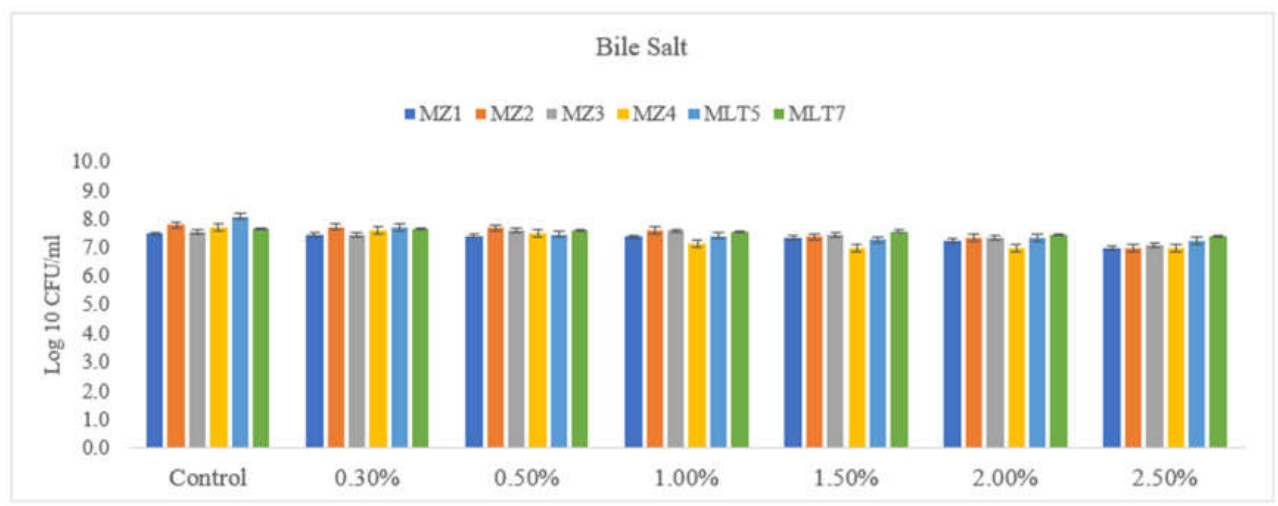

**Figure 6.** Slight decrease in the mean count of the test LAB isolates exposed to bile salt of different concentrations.

There was no significant difference ($p > 0.05$) among isolates at different bile salt concentrations.

### 3.4.6. Percentage Survival of the Isolates in the Varying Bile Salt Concentrations

The results of different concentrations of bile salt on percentage survival is presented in Table 4 below. The gastrointestinal tract's bile content fluctuates, so the isolates were tested for their ability to withstand bile salt at different concentration. All the different bile salt percentages did not show significant variation among the isolates ($p > 0.05$). However, numerically, MLT7 recorded the highest values for bile salt concentration at 1.0, 1.5, 2.0 and 2.5%.

**Table 4.** Effect of different % of bile salt on percentage survival of the isolates.

| Bile Salt Concentration (%) | MZ1 | MZ2 | MZ3 | MZ4 | MLT5 | MLT7 | SEM | *p*-Value |
|---|---|---|---|---|---|---|---|---|
| 0 | 28.9 | 44.4 | 35.6 | 37.8 | 24.4 | 33.3 | 10.1 | 0.789 |
| 0.3 | 20.0 | 40.0 | 28.9 | 33.3 | 37.8 | 33.3 | 10.8 | 0.818 |
| 0.5 | 22.22 | 35.56 | 28.89 | 22.22 | 26.67 | 31.11 | 8.11 | 0.831 |
| 1.0 | 17.78 | 31.11 | 26.67 | 11.11 | 20.00 | 26.67 | 6.48 | 0.338 |
| 1.5 | 17.78 | 17.78 | 20.00 | 6.67 | 13.33 | 26.67 | 3.39 | 0.025 |
| 2.0 | 13.33 | 15.55 | 15.55 | 6.67 | 15.55 | 20.00 | 2.27 | 0.082 |
| 2.5 | 6.67 | 6.67 | 8.89 | 6.67 | 13.33 | 17.78 | 2.03 | 0.009 |

Miaze. MLT—Pearl Millet; SEM—Standard Error of Mean. Means in the same row without superscripts are similar ($p > 0.05$).

### 3.4.7. Intestinal Fluid

The test LABs at 0 h recorded significant differences in mean count, but the values slightly decreased by about 2 log units at 2 h, where no difference was noted among the isolates ($p < 0.05$). However, varying increases in the mean concentration of the LABs were noticed at 4 h. In the case of 6 to 10 h, varying decreases in the mean concentration were also recorded but significant differences were observed at hour 4, 6, 8 and 10. The LAB strain representing MLT 5 demonstrated the highest tolerance, followed by MZ2 and MLT7, MZ3 and MZ4, with MZ1 representing the least. Figure 7 shows the tolerance capacity of the representative LAB strains of the various groups ($p < 0.05$).

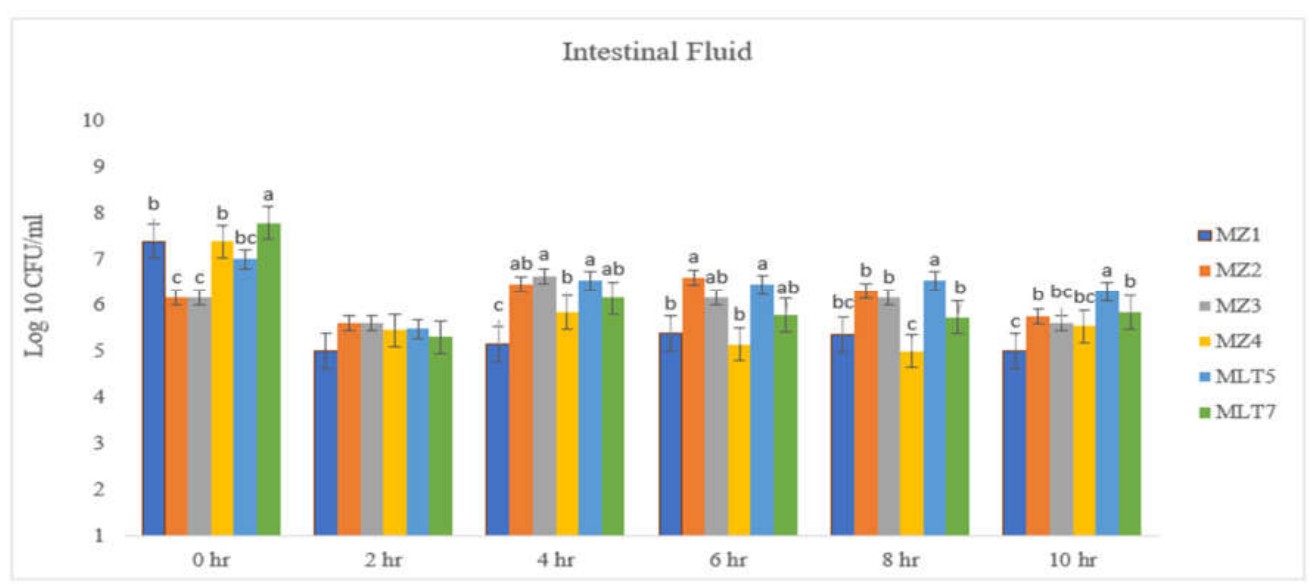

**Figure 7.** Decreasing patterns of the test LAB isolates exposed to intestinal fluid. [a, b, c] Means in the same row with different superscripts differ significantly ($p < 0.05$). Means in the same row without superscripts are similar ($p > 0.05$).

### 3.4.8. Percentage Survival of the Isolates against Intestinal Fluid

The percentage survivability of LAB isolates in the intestinal fluid was significantly affected for all periods except at 2 h. The percentage survival was significantly higher in MLT7, followed by MZ4 and MZ1, while MLT5, MZ2 and MZ3 recorded the least at 0 h. MZ3 did varied significant from MZ1 and MZ4 but was similar to MZ2, MLT5 and MLT7 at the 4th h. Additionally, MZ2 was significantly higher compared to MZ1, but both are similar to MZ3, MZ4, MLT5 and MLT7 at 6 h. Significant variation was observed at 8 h, where MLT5 was higher than LAB isolates MLT7, MZ4 and MZ1 but was similar to LAB isolates MZ2 and MZ3. At 10 h, MLT recorded the highest significant value compared to the other LAB isolates. MLT7, MZ2 and MZ3 were similar but significantly higher than MZ1 and MZ4 isolates, as shown in Table 5.

**Table 5.** Effect of intestinal fluid on percentage survival of the LAB isolates.

| | MZ1 | MZ2 | MZ3 | MZ4 | MLT5 | MLT7 | SEM | *p*-Value |
|---|---|---|---|---|---|---|---|---|
| Intestinal Fluid (%) | | | | | | | | |
| 0 h | 16.15 [b] | 1.44 [c] | 1.56 [c] | 17.19 [b] | 7.67 [bc] | 42.30 [a] | 2.20 | <0.01 |
| 2 h | 0.067 | 1.003 | 0.310 | 0.244 | 1.023 | 1.030 | 0.393 | 0.319 |
| 4 h | 0.089 [b] | 1.557 [ab] | 2.667 [a] | 0.410 [b] | 2.447 [a] | 0.890 [ab] | 0.386 | <0.01 |
| 6 h | 0.154 [b] | 1.957 [a] | 1.110 [ab] | 0.491 [ab] | 1.777 [ab] | 0.447 [ab] | 0.364 | 0.019 |
| 8 h | 0.102 [b] | 0.931 [ab] | 1.110 [ab] | 0.670 [b] | 2.0 [a] | 0.423 [b] | 0.246 | <0.01 |
| 10 h | 0.0603 [c] | 0.321 [bc] | 0.3 [bc] | 0.4010 [bc] | 1.33 [a] | 0.4867 [b] | 0.0808 | <0.01 |

Maize. MLT—pearl millet; SEM—standard error of mean. [a, b, c] Means in the same row with different superscripts differ significantly (*p* < 0.05). Means in the same row without superscripts are similar (*p* > 0.05).

### 3.4.9. Antimicrobial Resistance Patterns of the Test LABs

The test LAB isolates recorded high tolerance to the various antibiotics used after 18 to 24 h incubation except Erythromycin and Tetracycline which recorded a diameter zone of inhibition of 20 to 25 mm and 15 to 20 mm, respectively. Regrowth as shown on plate 2 was also observed after 24 h incubation. Figures 8 and 9 below shows the susceptibility of the test isolates to the two antibiotics and tolerance to the remaining six.

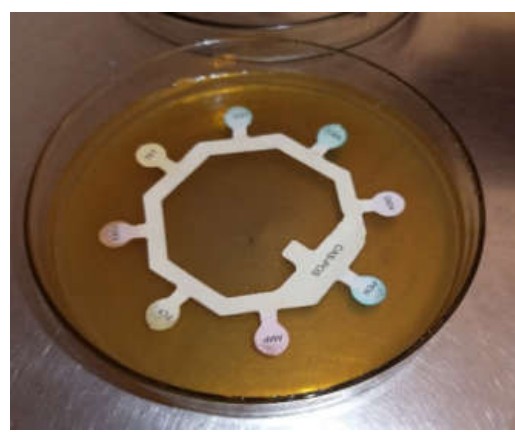

**Pate 1 Before**

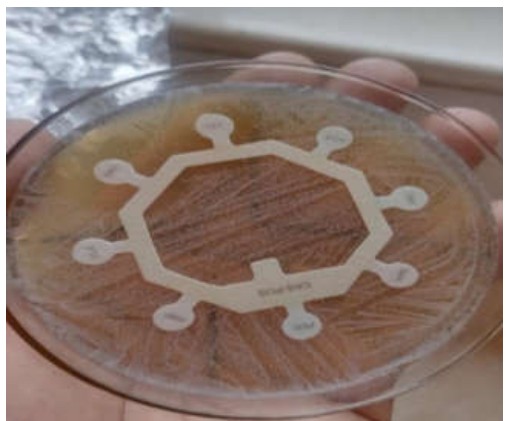

**Plate 2 After**

**Figure 8.** Resistant and susceptibility patterns of the LABs against the used antibiotics.

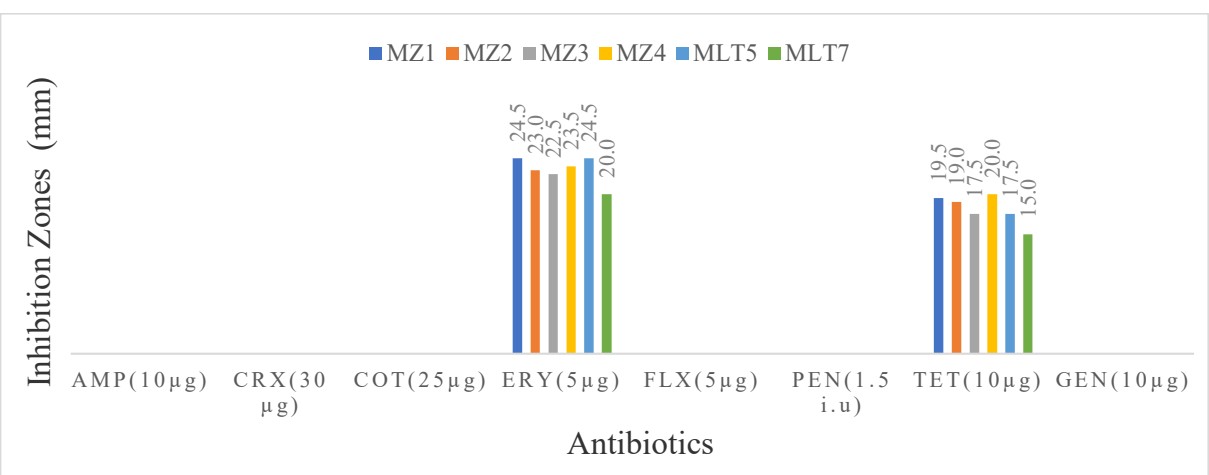

**Figure 9.** Resistance patterns of the test LAB isolates to the selected Gram-positive antibiotics.

3.4.10. Antimicrobial Activities against Pathogens

The antimicrobial activity of the chosen isolates was assessed against *Salmonella thyphimurium*, *Escherichia coli* (isolated from chicken) and *Escherichia coli* (isolated from pig), three indicator pathogens. The diameter zones of inhibition demonstrated that every isolate had antibacterial properties against the indicator pathogens, as shown in Figure 10.

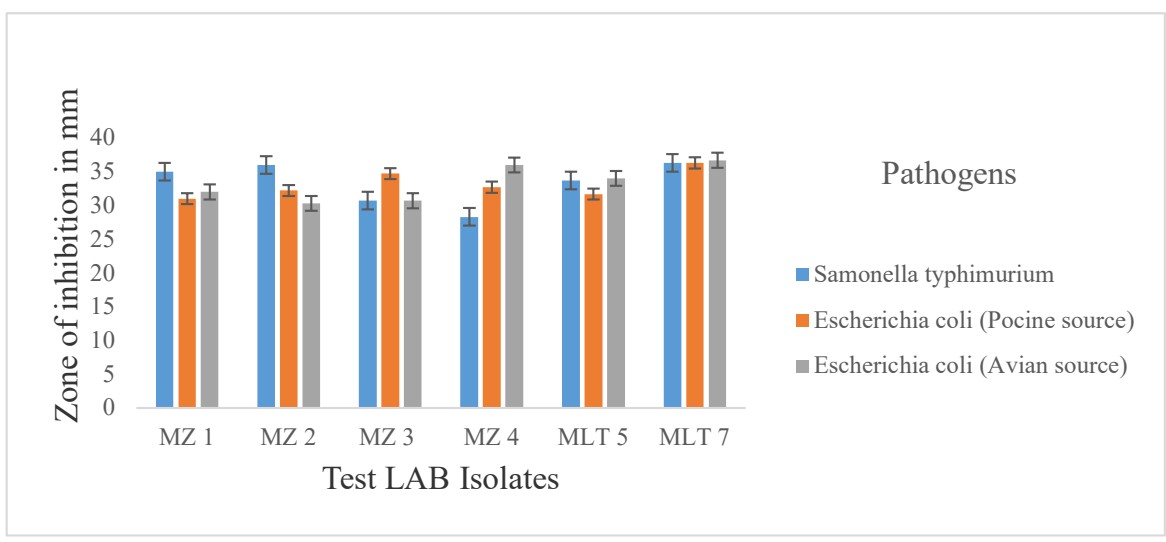

**Figure 10.** Antimicrobial activity of isolates against *Salmonella typhimurium*, *Escherichia coli* isolated from chicken and *Escherichia coli* isolated from pigs.

*3.5. Identification by 16S rRNA Sequencing*

Using 16S rRNA-amplified nucleotide sequences, the molecular identity of *Pediococcus pentosaceus* was verified. The DNA sequencing facility at Inqaba Biotech, Accra, Ghana, was utilized to sequence the PCR results. BLAST analysis was performed to search for the homology of 16S rRNA sequences that were present in the database using the 481 bp sequence that was acquired. Using MEGA 4.0, the unweighted pair group approach with arithmetic mean (UPGMA) was used to create the phylogenetic tree (Figure 11). The tree's nodes displayed the bootstrap values. The isolate (GT001) was identified as *Pediococcus pentosaceus* GT001 after it was discovered to have 100% similarity with some strains of *Pediococcus pentosaceus*.

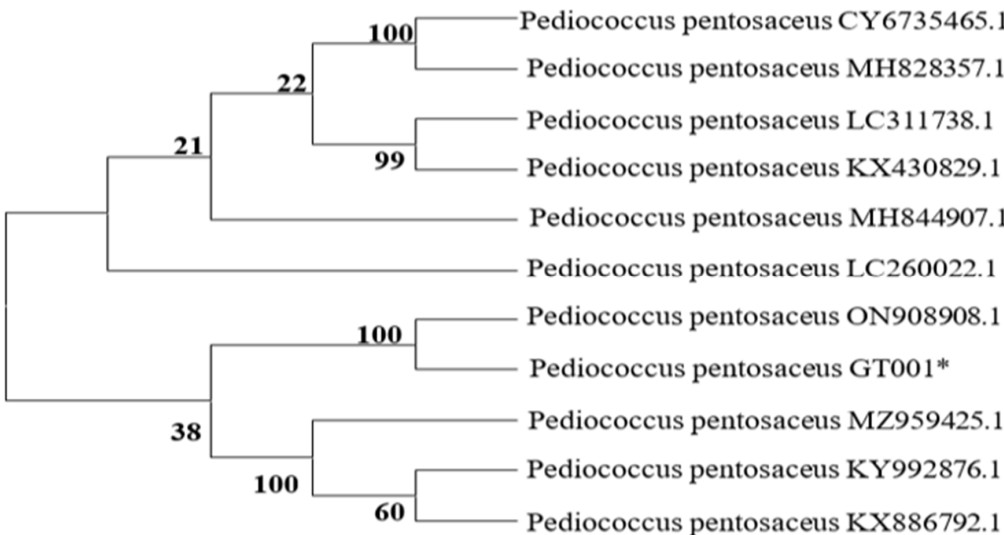

CTCAGGATGAACGCTGGCGGCGTGCCTAATACATGCAAGTCGAACGAACTTCCGTTAATTGATTATGACGTACTTGTACTGATTGAGATT
TTAACACGAAGTGAGTGGCGAACGGGTGAGTAACACGTGGGTAACCTGCCCAGAAGTAGGGGATAACACCTGGAAACAGATGCTAAT
ACCGTATAACAGAGAAAACCGCATGGTTTTCTTTTAAAAGATGGCTCTGCTATCACTTCTGGATGGACCCGCGGCGTATTAGCTAGTTGG
TGAGGTAAAGGCTCACCAAGGCAGTGATACGTAGCCGACCTGAGAGGGTAATCGGCCACATTGGGACTGAGACACGGCCCAGACTCCT
ACGGGAGGCAGCAGTAGGGAATCTTCCACAATGGACGCAAGTCTGATGGAGCAACGCCGCGTGAGTGAAGAAGGGTTTCGGCTCGTA
AAGCTCTGTTGTTAAAGAAGAACGTGGGTAAGAGTAACTGTTTACCCAGTGACGGTATTTAACCAGAAAGCCACGGCTAACTACGTGCC
AGCAGCCGCGGTAATACGTAGGTGGCAAGCGTTATCCGGATTTATTGGGCGTAAAGCGAGCGCAGGCGGTCTTTTAAGTCTAATGTGA
AAGCCTTCGGCTCAACCGAAGAAGTGCATTGGAAACTGGGAGACTTGAGTGCAGAAGAGGACAGTGGAACTCCATGTGTAGCGGTGA
AATGCGTAGATATATGGAAGAACACCAGTGGCGAAGGCGGCTGTCTGGTCTGCAACTGACGCTGAGGCTCGAAAGCATGGGTAGCGA
ACAGGATTAGATACCCTGGTAGTCCATGCCGTAAACGATGATTACTAAGTGTTGGAGGGTTTCCGCCCTTCAGTGCTGCAGCTAACGCAT
TAAGTAATCCGCCTGGGGAGTACGACCGCAAGGTTGAAACTCAAAAGAATTGACGGGGGCCCGCACAAGCGGTGGAGCATGTGGTTT
AATTCGAAGCTACGCGAAGAACCTTACCAGGTCTTGACATCTTCTGACAGTCTAAGAGATTAGAGGTTCCCTTCGGGGACAGAATGACA
GGTGGTGCATGGTTGTCGTCAGCTCGTGTCGTGAGATGTTGGGTTAAGTCCCGCAACGAGCGCAACCCTTATTACTAGTTGCCAGCATT
AAGTTGGGCACTCTAGTGAGACTGCCGGTGACAAACCGGAGGAAGGTGGGGACGACGTCAAATCATCATGCCCCTTATGACCTGGGCT
ACACACGTGCTACAATGGATGGTACAACGAGTCGCGAGACCGCGAGGTTAAGCTAATCTCTTAAAACCATTCTCAGTTCGGACTGTAGG
CTGCAACTCGCCTACACGAAGTCGGAATCGCTAGTAATCGCGGATCAGCATGCCGCGGTGAATACGTTCCCGGGCCTTGTACACACCGC
CCGTCACACCATGAGAGTTTGTAACACCCAAAGCCGGTGGGGTAACCTTTTAGGAGCTAGCCGTCTAAGGTGGGACAGATGATTAGGG
TGAAGTCGTA

**Figure 11.** Phylogenetic tree of *Pediococcus pentosaceus* GT001 (indicated by asterisk *).

## 4. Discussion

### 4.1. Isolation and Identification of the Test LAB Isolates

The purpose of this study was to isolate, characterize and investigate the probiotic potential of *pediococcus* sp. in fermented cereal (maize and pearl millet) dough. Cereals are different and accessible sources of lactic acid bacteria with the potential for the development of probiotics, and probiotic strains may also work better in an environment similar to their sources of isolation. The media used for LAB isolation was MRS agar (Oxoid CM361) and MRS broth (Oxoid CM359). The isolates were subcultured, and the colonies were purified via continuous streaking on MRS agar plates. Phenotypic characterization was determined on each isolate according to De Vries et al. [17], and LAB species with similar phenotypic characteristics were grouped and identified using matrix-assisted laser desorption ionization-time of flight mass spectrometry (MALDI-TOF MS). This identification tool was chosen because of the research conducted by Ozaslan et al. [18] reported MALDI-TOF MS to be an identification tool that can be used for the rapid identification of lactic acid bacteria associated with fermented products, and its mechanism of identification is closely associated with that of the RNA gene 16S rDNA sequence identification method [19].

*4.2. Acid and Gastric Juice Tolerance*

The ability of prospective probiotic bacteria to survive passage through the gastrointestinal tracts to colonize the intestines when administered orally is key to the use. LABs intended for probiotic use must possess resistance to low pH, gastric juice and bile salt since these are powerful barriers that face bacteria when entering the intestinal tract [12]. Most microorganisms generally show high sensitivity to pH levels below 3.0 [20], meaning most probiotics find it difficult to make it through into the small intestine in larger quantities due to the low pH of gastric juice, limiting their effectiveness in most functional foods [21]. The acid tolerance assay results obtained from this study indicated a significant percentage survival for all test isolates of the *Pediococcus* strains exposed to pH 3 and pH 4 for 3 h. In terms of pH 2 and pH 1, the isolates were recorded as having reduced percentage survival, which is in line with the results reported by Rönkä et al. [20]. Isolate representing group MZ4 recorded the highest percentage survival, followed by that of MLT 5 and MZ2 but not as significant as *Lactobacillus brevis* strains GRL1 and GRL62 recorded by Rönkä et al. [20]. According to the FAO/WHO, [22] food promoted with claims of health benefits in the form of probiotics must have viable culture cells of at least $10^6$–$10^7$ CFU/g and, if ingested, the cells must also survive passage in adequate quantity.

LAB strains from various sources have different viabilities in terms of pH according to Takanashi et al. [23]. However, each LAB strain adapts to its particular environment, as indicated by Succi et al. [24]. However, most probiotic bacteria in use belong to the genera *Lactobacillus*, *Bifidobacterium*, *Streptococcus*, *Lactococcus* and *Enterococcus* [25,26]. These *Pediococcus* strains isolated from fermenting maize and pearl millet dough and subjected to stressful conditions of pH 2 and pH 1 for 3 h, similar to those isolated from fermented vegetables [27], indicated that these *Pediococcus* strains, just like other LABs, such as *Lactobacillus rhamnosus*, survived acidity as low as 2.5 for about 4 h [28] due to the fact that they possess a constant gradient exists between their extracellular and cytoplasmic pH [29], which enable them to adapt during the maize or millet dough fermentation, and this may also enable these dough fermented *Pediococcus isolates* to survive the gastric juice in the gastrointestinal track of broiler chickens.

*4.3. Bile Salt Tolerance*

In the case of bile salt, Boke et al. [30] indicated that LAB intended for probiotic use must be able to tolerate bile salt at a concentration level of 0.15–0.3%, a reference considered to be most appropriate for selecting probiotic bacteria. It is estimated that the average intestinal bile content is 0.3% $w/v$, and a 4 h stay is advised [25]. Even though bile tolerance is generally strain-dependent according to Adefisoye et al. [29], Gram-positive bacteria are more susceptible to bile salts, and it is not possible to identify a particular species of bacterium that is more resistant to bile salts than others since bile tolerance is strain-specific, according to Mishra and Prasad [31]. Variations in pH, temperature and other environmental settings can either increase or decrease a bacteria's resistance to bile salts [32]. The tolerance of probiotics to high bile concentrations can also be improved by pre-exposing them to a low concentration of bile salts for a period of time [33].

In this study, the results obtained for all the test isolates indicated significant tolerance to bile salt, similar to what Takanashi et al. [23] recorded, indicating that *Pediococcus* is one of the LAB strains possessing the ability to deconjugate bile salt, suppressing its reabsorption, which means they will be effective in reducing cholesterol in the serum of broiler chickens [12] by inhibiting cholesterol absorption. Additionally, being a probiotic candidate, they will have a tendency to hydrolyze bile salt in the intestinal lumen of broiler chickens, leading to a reduced breakdown of lipids [14]. Secondly, in a molecular study conducted on *Pediococcus acidilactici* strains [34], it was realized that LABs tolerant to bile salts possess several hydrolase genes such as groEL, dltD, clpL and the bsh gene, which are linked to bile salt and acid.

*4.4. Intestinal Fluid Tolerance*

Tolerance to intestinal fluid is another vital criterion when selecting probiotic bacteria [35]. The ability of a bacterium to adhere to the epithelial cells of the intestine of the host and colonize it via antagonistic activity against its competitive pathogens depends on its ability to withstand the intestinal fluid. *Lactobacillus* are more tolerant to intestinal fluid since they are part of the natural intestinal microbiota [36]. Perdigón et al. [37] also recorded *Enterococcus*, *Lactococcus* and *Pediococcus* as LABs tolerant to intestinal fluid. However, all the *Pediococcus* strains used in this study demonstrated strong tolerance to the intestinal fluid even at the 10 h mark, indicating similarities to what Perdigón et al. [37] reported.

*4.5. Antibiotic Tolerance and Properties of the Test LABs*

Resistance to antibiotics by probiotic bacteria is very important because it allows them to withstand antibiotic treatments [38]. However, the expected potential of probiotic bacteria to transfer this resistance to pathogenic bacteria is not acceptable [39]. The results obtained in this study indicated the resistance of all the *Pediococcus* strains against six of the antibiotics used except Erythromycin and Tetracycline, wherein regrowth occurred after 24 h of incubation. The results were similar to those reported by Cao et al. [38] for Tetracycline but contrary to that of gentamycin. In terms of the antimicrobial activity of the *Pediococcus* strains against the selected pathogens, all of the LAB isolates exhibited antimicrobial activity against *Salmonella typhimurium*, *E. coli* (chicken) and *E. coli* (pig) by producing zones of inhibition ranging from 28 mm to 36 mm in diameter, which is an indication that the *pediococcus* strains were able to produce antimicrobial products, including lactic and acetic acid, that the three indicator pathogens (*E. coli* P, *E. coli* C and *Salmonella typhimurium*) were susceptible to [40].

## 5. Conclusions

The exploitation of antibiotics in animal production has led to antibiotic resistance and a possible decrease in the effectiveness of both animals and humans treatment, and the use of lactic acid bacteria (LAB) such as *Pediococcus* strains in the form of probiotics, will play an important role in contributing to an increasing growth performance in livestock management and animal health.

After a methodical in vitro screening process, strain MLT7 isolated from pearl millet and identified as *Pediococcus pentosaceus*, both with MALDI-TOF MS and 16S rRNA sequences, was ultimately chosen due to its encouraging probiotic qualities and lack of harmful traits. This strain was selected for in vivo research to clarify its potential health advantages and for use as a novel probiotic strain in supplements for animal feed as part of a Ph.D. project acceded on 1 December 2023 (https://doi.org/10.3390/ani13233724).

**Author Contributions:** G.Z.B.: investigation, data collection, methodology, statistical analysis and writing—original draft. L.A., P.A., A.D.O.-O. and T.N.: investigation, methodology, data collection and writing—review and editing. B.B.B., F.D., K.O.A. and T.M.M.: statistical analysis, review and editing. Z.T.: investigation, project administration, funding acquisition, supervision and writing—reviewing and editing. All authors have read and agreed to the published version of the manuscript.

**Funding:** This research was funded by the Fundamental Research Funds for the National Key R&D Program of China (2022YFD1300501-3), and the Natural Science Foundation Project of Chongqing (cstc2021jcyj-msxmX0966).

**Institutional Review Board Statement:** The Council for Scientific and Industrial Research (CSIR) Institutional Animal Care and Use Committee (IACUC), Ghana (RPN 008/CSIR-IACUC/2022, approval date: 21 July 2023) approved the protocol, and this research was conducted in accordance with its guidelines.

**Informed Consent Statement:** Not applicable.

**Data Availability Statement:** Upon request, the corresponding author may supply the data.

**Conflicts of Interest:** The authors declare no conflict of interest.

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
