# Peer review of "In Vitro Evaluation of Gastrointestinal Stability of Pediococcus pentosaceus Isolated from Fermented Maize and Pearl Millet for Possible Novel Chicken Probiotic Development"

_2036-7481, doi:10.3390/microbiolres15020051_

Round 1

Reviewer 1 Report

Comments and Suggestions for Authors

The work is devoted to the screening of potential probiotics from the point of view of their survival in conditions emitting the gastrointestinal tract, as well as their resistance to antimicrobial drugs. This topic is relevant because the use of antibiotics in animal agriculture has led to antibiotic resistance and the possible reduction in the effectiveness of medical treatment.

There are the following comments to the article:

- Section 3.3 does not provide specific results of studies using the MALDI-TOF mass-spectrometry method on the basis of which the isolates were differentiated. At the same time, in the discussion, the authors provide numerous arguments for the use of this technique (lines 497-515). Authors should provide information about the identified components on the basis of which the differentiation was made.

- Sometimes authors use terminology that raises questions. For example, the authors use the term “percentage survival rate”. But rate is a quantity that is equal to the increase (decrease) of some value per unit of time. This is not in the definition of “Rate of survival” (line 196). The authors also use the term pH concentration, instead of just pH. Authors should use commonly used terms.

- Authors need to check the text for typos. For example: “log 10” instead of log10 (or log10), both “survival percentage” and “percentage survival” (one or the other should be used), etc.

Author Response

General comment: The work is devoted to the screening of potential probiotics from the point of view of their survival in conditions emitting the gastrointestinal tract, as well as their resistance to antimicrobial drugs. This topic is relevant because the use of antibiotics in animal agriculture has led to antibiotic resistance and the possible reduction in the effectiveness of medical treatment.

Authors response: We appreciate the reviewer's insightful remarks and recommendations. We have implemented extensive changes based on the recommendations. A revised manuscript with the detailed modifications have been highlighted.

Comment 1: Section 3.3 does not provide specific results of studies using the MALDI-TOF mass-spectrometry method on the basis of which the isolates were differentiated. At the same time, in the discussion, the authors provide numerous arguments for the use of this technique (lines 497-515). Authors should provide information about the identified components on the basis of which the differentiation was made.

Authors response: According to your suggestion, the result of MALDI-TOF MS that serves as basis in differentiating the isolates have been included. (Please see line 302 and 303).

Comment 2: Sometimes authors use terminology that raises questions. For example, the authors use the term “percentage survival rate”. But rate is a quantity that is equal to the increase (decrease) of some value per unit of time. This is not in the definition of “Rate of survival” (line 196). The authors also use the term pH concentration, instead of just pH. Authors should use commonly used terms.

Authors response: All percentage survival rate and rate of survival have been modified to percentage survival as suggested. All pH concentration has been changed to pH as suggested as well. (Kindly check lines 199, 201, 332, 333,335, 336, 342, 346, 351, 354, 390,391, 392,395,400, 419,420,428,449,450,451,460,524,525 and 527 for percentage survival and 194,195,198 and 331 for pH).

Comment 3: Authors need to check the text for typos. For example: “log 10” instead of log10 (or log10), both “survival percentage” and “percentage survival” (one or the other should be used), etc.

Authors response: All log 10 have been changed to log10. (Please check lines 311, 312, 376, 377, 379, 380,411).

Reviewer 2 Report

Comments and Suggestions for Authors

Please have the attached file

Author Response

General comment: This article presents the resistance toward acid, bile salt and Intestinal fluid as well as antimicrobial activity for 8 Pediococcus spp strains isolated from fermented Maize and Pearl millet and expected to apply the isolates as Novel Chicken Probiotic. The weakness of this article is that the isolated Pediococcus spp were only assessed on their resistance toward gastrointestinal digestion and antibiotic. The discussion section of the text is too weak to convince readers that flakes isolated from corn and pearl millet dough are added to animal feed. cocci strains are superior to adding commercially available lactic acid bacteria.

Authors response: We appreciate the reviewer's insightful remarks and recommendations. We have implemented extensive changes based on the recommendations. A revised manuscript with the detailed modifications have been highlighted.

Comment 1: Suggest the title of this article should be modified. As stated in this article "The purpose of the study was to isolate, characterize and study the probiotic properties of Pediococcus sp.", however, the probiotic properties of the isolated strains were not well evaluated. Being a probiotic not only survives in gastrointestinal digestion, it is also important to promote health.

Authors response: As suggested, the tittle of the article has been modified. Thank you. (kindly check line 2-4).  

Comment 2: Line 31-38 should be streamlined.

Authors response: Line 31-38 has been streamlined. (Please check line 31-43)

Comment 3: 2.3.6. Antimicrobial activity against three selected pathogens. In this section, macfarlane standard was used to compare the suspension of bacteria with and without the present of isolated strains. What kind of medium was used to culture each pathogen?

Authors response: The medium used for the selected pathogens was Nutrient agar and it is stated in 2.3.6. (Kindle check line 266)

Comment 4: Line 289, Based on the phenotypic characteristics, the isolates were placed into eleven (8) groups and coded MZl, MZ2, MZ3, MZ4, MLTS, MLT 6, MLT7 and MLT8. The results

Authors response: It has been modified. Thank you. (Please check line 296).

Comment 5: Line 312, However, significant differences were noted at pH 3, 2 ans and 1 among the … Please delete redundant words -“ans”.

Authors response: It has been deleted. Thank you. (Kindly check line 321)

Comment 6: Table 3, All statistical symbols must be superscripted.

Authors response: All statistical symbols have been made superscript. Thank you. (Kindly check line 401).

Comment 7: Table 3 and 5. The unit of calculation of survival rate was percentage?

Authors response: Yes please and the changes have been effected. Thank you. (Kindly check line 401 and 462)

Comment 8: Figure 3. Please use the same Y-axis scale for each subplot.

Authors response: The same Y-axis scale has been used for all subplot. Thank you. (Please check figure 4, line 324).

Comment 9: Figure 4. Please add a description for the statistical symbols.

Authors response: Statistical symbols have been described. (kindly check figure 5, line 388 and 389)

Comment 10: Figure 5, Please add a note: there is no significant difference (P > 0.05) among groups at the same bile salt concentration.

Authors response: The note has been added. (Please check figure 6, line 418)

Comment 11: Figure 6. Please add a description for the statistical symbols. All statistical symbols must be superscripted

Authors response: Statistical symbols have been described. (Please check figure 7, line 447 and 448).

Comment 12: Line 493-515. The method of using MALDI-TOF MS to identify bacterial species has been developed for more than 10 years. It is recommended that this paragraph needs to be streamlined.

Authors response: The discussion on MALDI-TOF MS has been streamlined. (Please check line 508 to 515).

Comment 13: Line 524-539. This part almost duplicates the "Results" section should be streamlined or rewritten.

Authors response: Thank you. That section has been streamlined. (Kindly check line 524 to 532)

Comment 14: Please provide a stronger argument highlighting the potential of isolated strains as probiotics for chickens.

Authors response: There is an article highlighting the potential of the selected isolated strain as probiotic for chickens and the doi has been stated. (Please check line 605).

Round 2

Reviewer 2 Report

Comments and Suggestions for Authors

Accept this version